# A Study on Deep Learning-Based Fault Diagnosis and Classification for Marine Engine System Auxiliary Equipment

**Jeong-yeong Kim** [1,2,†], **Tae-hyun Lee** [1,†], **Song-ho Lee** [1], **Jong-jik Lee** [1], **Won-kyun Lee** [2], **Yong-jin Kim** [1,*] **and Jong-won Park** [1,*]

1 Department of Reliability Assessment, Korea Institute of Machinery and Materials, Daejeon 34103, Korea; jykim8792@kimm.re.kr (J.-y.K.); thlee07@kimm.re.kr (T.-h.L.); dlrm741@kimm.re.kr (S.-h.L.); ljjik@kimm.re.kr (J.-j.L.)
2 School of Mechanical Engineering, Chungnam National University, Daejeon 34134, Korea; wklee@cnu.ac.kr
* Correspondence: yjkim2014@kimm.re.kr (Y.-j.K.); jwpark@kimm.re.kr (J.-w.P.)
† These authors contributed equally to this work.

**Abstract:** Maritime autonomous surface ships (MASS) are proposed as a future technology of the maritime industry. One of the key technologies for the development of MASS is condition-based maintenance (CBM) based on prognostics and health management (PHM). The CBM technology can be used for early detection of abnormalities based on the database and for a prediction of the fault occurring in the future. However, this technology has a problem that requires a high-quality database that reproduces the operation state of the actual ships and quantitatively and systematically indicates the characteristics for the various fault state of the device. To solve this problem, this paper presents a study on the development method of the fault database based on the reliability. Firstly, the reliability analysis of the target device was performed to select five types of the core fault modes. After that, a fault simulation scenario that defined the fault simulation test methodology was drawn. A land-based testbed was built for the fault simulation test. The fault simulation database was developed with a total of 109 sets through the fault simulation test. Additionally, a fault classification algorithm based on deep learning is proposed. The classification performance was evaluated with a confusion matrix. The developed database will be expected to serve as the basis for the development CBM technology of MASS in the future.

**Keywords:** condition-based maintenance (CBM); fault diagnosis; 1D CNN; fault simulation database

## 1. Introduction

Maritime autonomous surface ships (MASS) are attracting attention as the future technologies in the maritime industry. One of the core technologies for the development of MASS is the self-diagnosis ship technology. The self-diagnosis ship is a technology based on condition-based maintenance (CBM) that can improve the safety and efficiency by performing autonomous decision-making using an artificial intelligence (AI) system based on the database. In addition, this technology can replace the work that depended on the experiential decisions of sailors [1].

Recently, the CBM technology has been actively studied in most industries such as nuclear and thermal power plants. It is expected to improve health management technology and reduce the maintenance costs by minimizing the replacement of parts through predictive maintenance (PM) [2]. However, maintenance technology of ships has depended on breakdown maintenance (BM) technologies that are repaired in case of malfunction and time-based maintenance (TBM) technology that is repaired after use for a certain period of time. Therefore, maintenance technology of ships has the problem of high-costs due to frequent downtime and replacement of parts [2]. In order to solve this problem, the CBM technology of ships that monitors the state in real time and detects the fault of a system should be developed. In addition, the development of CBM technology



can establish efficient measures about the maintenance to reduce unnecessary maintenance costs and downtime and to increase the safety and reliability of the system.

The first step for the development of CBM technology is acquire the fault simulation database of various types [3]. However, study on the development of the database is lacking because the maintenance technologies of the ships such as BM and TBM were performed. In addition, there is a problem in that it takes a long time to acquire various databases of the ships. To solve this problem, the fault simulation database will be developed through land tests. The second step for technology development is to diagnose and predict the faults based on AI algorithms. Recently, a study on a fault diagnosis based on AI is being actively performed [4]. In addition, the fault diagnosis system of existing ships has been developed so that engine manufacturers can monitor and diagnosis their own engine faults. This system is limited as systems of large ships consider the working environment and user environment of the ship [5].

A fault diagnosis system for MASS must be developed not only in the engine but also in the most auxiliary devices surrounding the engine system. The supply pump is the auxiliary device most closely installed with the engine and is a core device for the operation and fuel supply of the engine [6,7]. Accordingly, a fault of the supply pump may cause a critical fault of the engine system, which is core to the safety of the ship's operation and on-time arrival and departure [8]. Therefore, it is obvious that the CBM development of the fuel supply pump is a technology that improves the safety and efficiency of MASS.

In this paper, in order to develop the fault simulation database based on reliability, we propose the experimental study of the fuel supply pump. First, a fault case and reliability analysis of the target device was performed to select the core fault modes. After that, the fault simulation scenario that defined the fault simulation test methodology was drawn. A land-based testbed was built for the fault simulation test. By using a land-based testbed, the fault simulation database was developed. Additionally, the algorithm model based on deep learning was proposed for the fault diagnosis and classification. The performance of the algorithm was evaluated with a multi-class confusion matrix.

## 2. Reliability Analysis of Screw Pump

### 2.1. Target Device

The engine system of the ships was composed as various auxiliary devices such as the heaters, the purifiers, and the pumps. The fuel supply pump is most closely installed with the engine. In addition, it is a core device of the engine system that supplies the fuel to the engine quickly and stably [6]. The target device to use in this study was selected as the three-axis screw pump from KRAL co., ltd., Gyeonggi-do, Korea, which has a similar function and configuration of the supply pump used in actual ships. The structure of the pump is shown in Figure 1.

The three-axis screw pump consists with the main rotor and two idlers and jaw coupling that transmits power of the motor. The maximum speed of the pump rotates at a speed of 1770 RPM, and it stably supplies the fuel of 100 L for a minute despite rotating at a high speed.

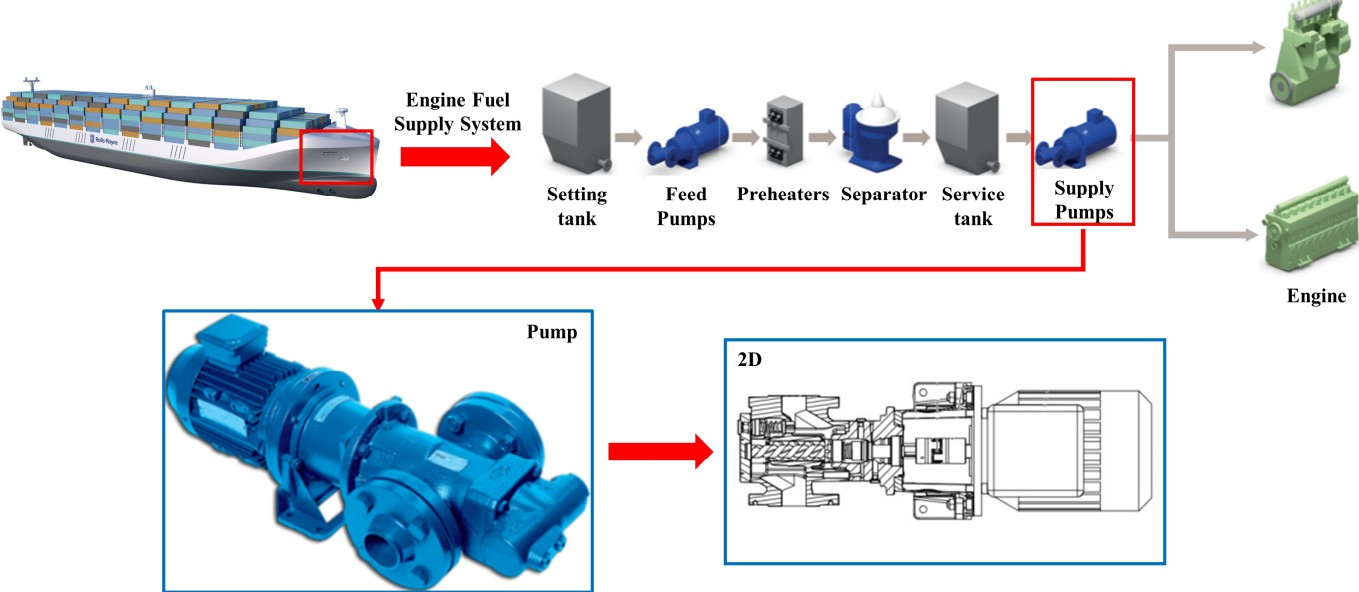

**Figure 1.** The 3-axis screw pump of target device. Reprinted/adapted with permission from Refs. [6,9].

## 2.2. Reliability Analysis

The reliability analysis was performed to quantitatively represent the core fault modes of the target device. Firstly, the fault case analysis was performed to draw the risk, operating characteristics, performance requirements, and fault rate accounting to fault of the fuel supply pump using the ship equipment fault case collection [10] and offshore reliability data [11]. After that, the three techniques of the reliability analysis used: fault modes, effects, and criticality analysis (FMECA), criticality matrix analysis (CMA), and fault tree analysis (FTA). In addition, the risk priorities were evaluated using the potential fault types, causes, and mechanisms of fault and methods of detection and management. The five types of the core fault modes such as bearing lubrication and wear, coupling elastomer wear, mechanical seal degradation, misalignment, and cavitation were selected as the results of the reliability analysis. The results of reliability analysis are shown in Table 1.

**Table 1.** Fault modes, effects and criticality analysis (FMECA) for 5 types fault modes of pump.

| Main Parts | Fault Modes | Fault Causes and Mechanisms | Fault Effect | Criticality | | |
|---|---|---|---|---|---|---|
| | | | | Occurrence | Severity | Detection |
| Motor/Pump bearing | Sticking of bearing | Particle injection, Poor lubrication | Occurrence of vibration and noise | 5 | 4 | 2 |
| Mechanical seal | Leakage | Deterioration, Poor lubrication | Reduced Efficiency | 5 | 4 | 2 |
| Jaw coupling | Elastomer wear | Overload, Overheating | Increase vibration | 5 | 5 | 3 |
| | Coupling wear | Misalignment | Increase vibration | 4 | 5 | 2 |
| Rotor | Rotor crack | Cavitation | Occurrence of vibration and noise | 5 | 5 | 2 |

## 2.3. Fault Simulation Scenario

The fault simulation scenario that defined an experimental methodology based on several prior studies for the fault simulation tests of five types of the fault modes was developed. In addition, it presented the fault simulation diagram to visualize the test methodology. The fault simulation scenario is shown in Table 2.

**Table 2.** Fault simulation scenario for 5 types of the fault modes of pump.

| Fault Modes | Motor and Pump-Bearing Lubrication/Particle Injection | Mechanical Seal Degradation | Jaw Coupling Elastomer Wear | Misalignment | Cavitation |
|---|---|---|---|---|---|
| Fault test plan | Complete remove of lubricant Particle injection (0.5 g) | 120 °C, 90 h degradation | Normal: 0% Remove1: 25% Remove2: 50% Remove3: 75% | Angular: max. 0.9° Lateral: max. 0.25 mm Offset: max. 4 mm | Outlet pressure change |
| Fault simulation diagram | | | | | |

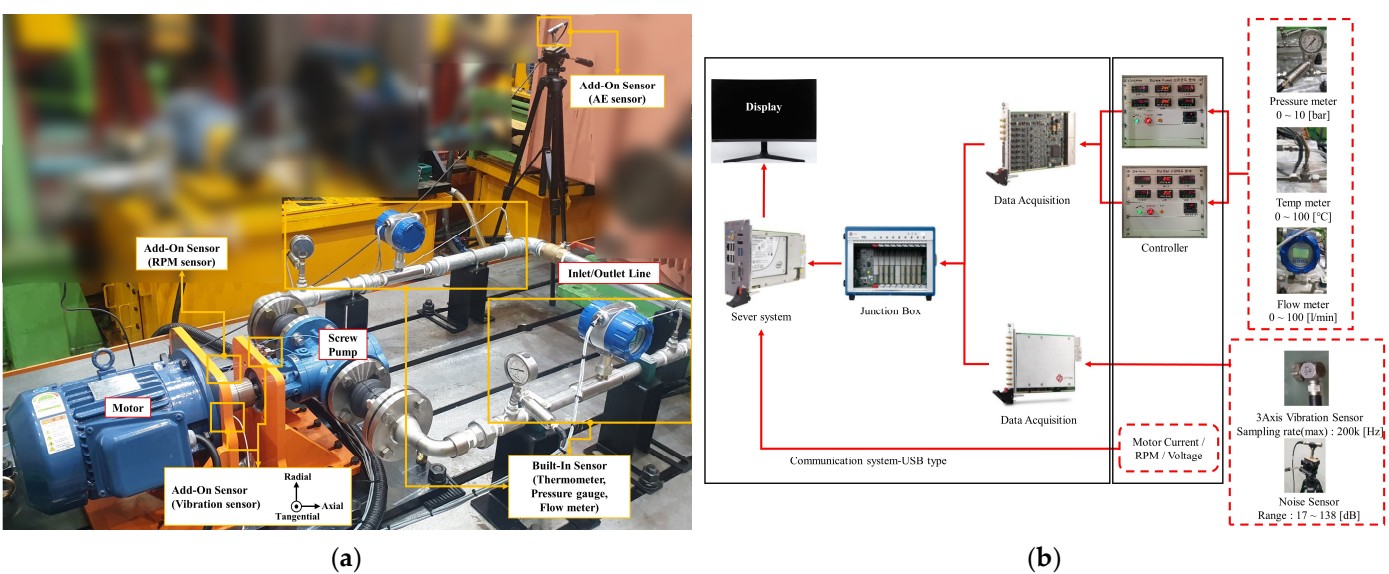

The bearing lubrication fault test completely removed the grease, and the particle injection test was performed by injecting 0.5 g of the particle. The jaw coupling elastomer wear test was simulated in a degradation test while removing the wings of the normal elastomer. The mechanical seal was degraded at 120 °C for 90 h for fault simulation. In the case of the misalignment test, the fault simulation test was applied as the maximum change of the angular, lateral, and offset change in the design specification standard. Finally, the cavitation test was performed by applying the artificial cavitation according to the change in outlet pressure.

## 3. Development of the Fault Simulation Database

### 3.1. Experimental Testbed

Study on the development of the fault simulation database was limited in terms of time and finance. Therefore, the database was to be acquired through land tests. A land-based testbed was built for the fault simulation test. The engine system of the actual ships was built as the combined system of various auxiliary devices such as the purifiers, the fuel tanks, and the fuel supply pumps. But the land-based testbed was built as an individual system to monitor the functional parts of the fuel supply pump. The structure of the land-based testbed is shown in Figure 2.

**(a)**                                                                    **(b)**

**Figure 2.** A land-based testbed for the fault simulation test: (**a**) Testbed facility picture; (**b**) Multi-parameter monitoring system.

A real-time status monitoring and data acquisition system was built using the C# S/W from JYTEK inc., Gyeonggi-do, Korea. In addition, the Amadeus S/W from FAMTECH Co., Ltd., Gyeongnam, Korea, monitored and analyzed the characteristic factors in the high-frequency range.

The measurement locations of the built-in, vibration, and noise sensors were selected according to KS Standards [12,13]. The data sampling rates were 1 kHz, 25.6 kHz, and 51.2 kHz, and it was acquired to observe in the maximum frequency range of the data. In addition, the three-axis accelerometer from PCB piezotronics, inc., KISTLER co., ltd., Walden Ave, NY, USA, was used to acquire the various data.

### 3.2. Fault Simulation Test

The fault simulation test was performed with reference to the test methods through several previous studies [14–25]. By using the test methodology, a specimen for the fault simulation test was prepared. After that, the fault specimen was reinstalled to monitor the performance according to the fault effect. The methodology for each fault simulation test is listed in Sections 3.2.1–3.2.5, and the results of the fault simulation test are shown in Figure 3.

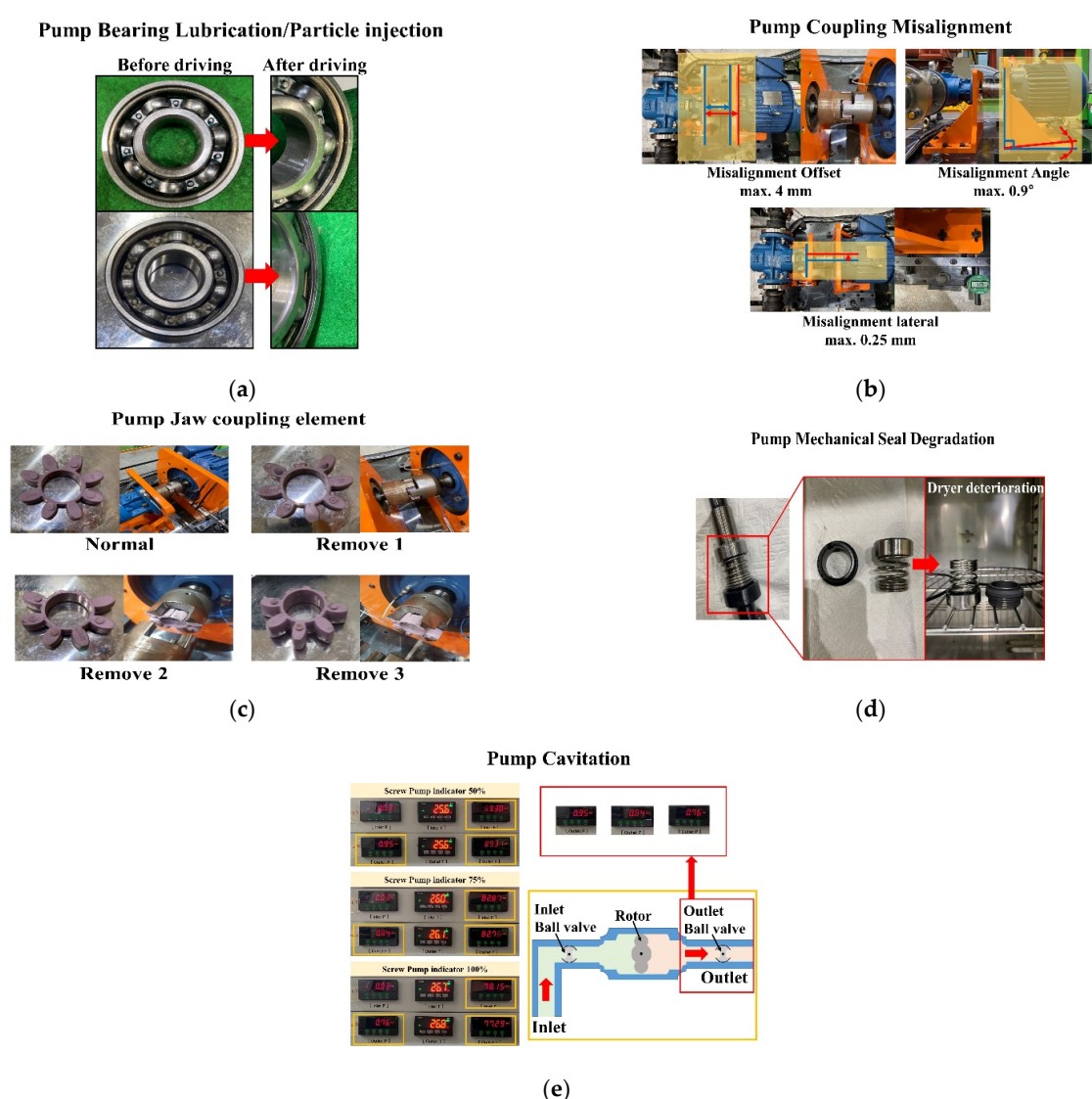

**Figure 3.** The picture for the process of performing of the fault simulation test for the fault modes of pump: (**a**) Motor and pump bearing lubrication and particle injection test; (**b**) Misalignment test; (**c**) Jaw coupling element test; (**d**) Mechanical seal degradation test; and (**e**) Cavitation test.

### 3.2.1. Pump/Motor Bearing

Figure 3a shows the results of the bearing fault test. The fault simulation was performed by lubrication fault and particle injection tests. The lubrication fault test was used to simulate the evaporation of the lubricating oil of the bearing grease, and the fault was simulated by completely removing grease from the bearing using a lubricating remover. The particle injection test was used to simulate wear fault of the inner and outer rings due to the particle entering the bearing, and the fault was simulated by injecting 0.5 g of 300 μm particles.

### 3.2.2. Misalignment

Figure 3b shows the process of performing the fault simulation test for the misalignment test. The misalignment test was performed to observe the effect on coupling when misalignment occurred. The fault simulation test was performed by applying the maximum change based on the design specification of the device.

### 3.2.3. Jaw Coupling

Figure 3c shows the results of the jaw coupling elastomer wear test. In the fault simulation test, the elastomer wing was removed to simulate the fault for elastomer wear that occurs during overheating of the coupling. The elastomer wing was removed up to three to simulate the faults occurring according to the deterioration step.

### 3.2.4. Mechanical Seal

Figure 3d shows the process of performing a mechanical seal deterioration test. Mechanical seal deterioration was simulated by thermal aging due to friction of the seal during pump operation. Thermal aging was performed at 120 °C for 90 h using a heat dryer. After thermal aging, the mechanical seal was reinstalled on the pump and operated to observe the functional effect of the pump.

### 3.2.5. Cavitation

Figure 3e shows the cavitation simulation test with indicators and diagrams. This was intended to simulate a cavitation fault caused by a pressure difference in the service tank or particle in the pipe. In the fault simulation test, cavitation was simulated by an artificial fault by reducing the outlet pressure by up to 20% above the normal pressure.

### 3.3. Fault Simulation Database

In order to perform the fault diagnosis, it is important to collect a high-quality database [26]. The database was acquired through the fault simulation test. It was saved as a csv. file with a tag number designated. Table 3 shows the acquired database status. Through the normal test and fault simulation tests and additional tests were data acquired.

The database was acquired as at least three sets for each fault mode, normal data were acquired as five sets. The motor and pump bearing lubrication fault and particle injection test data were acquired as a total of 20 sets, as each of 5 sets. For the jaw coupling fault simulation test, the misalignment test and elastomer wear test were performed. In case of the misalignment test, the test data were acquired as a total of 32 sets as offset, angular, and lateral tests. In the case of the elastomer wear test, the test data were acquired as a total of 18 sets by removing up to three elastomer wings. For the mechanical seal test, a degradation test was performed, and the test data were acquired as a total of five sets. The cavitation test was performed by controlling the outlet pressure, and the test data were acquired as a total of four sets. Through additional tests such as the bearing life test and the complex fault test, a total of 25 sets were acquired. Through the fault simulation test, a total of 109 sets were acquired, and a database was developed as a csv. file with a capacity of 500 GB.

**Table 3.** Database development status of pump.

| Device | Components | Fault Simulation Test | | Data Acquisition (Set) |
|---|---|---|---|---|
| Pump | Normal | Normal | | 5 |
| | Motor Bearing | Lubrication | | 5 |
| | | Particle injection | | 5 |
| | Pump Bearing | Lubrication | | 5 |
| | | Particle injection | | 5 |
| | Jaw Coupling | Misalignment | Offset | 20 |
| | | | Angular | 6 |
| | | | Lateral | 6 |
| | | Elastomer | Remove1 | 8 |
| | | | Remove2 | 5 |
| | | | Remove3 | 5 |
| | Mechanical Seal | Mechanical Seal Degradation | | 5 |
| | Rotor | Cavitation | | 4 |
| | Option test | Bearing life test | | 25 |
| | | Complex fault test | | |
| | Total | Built-in data | | 109 |
| | | Add-on data | | |
| | | Noise data | | |

*3.4. Data Analysis*

The causes of malfunction of rotating equipment are mostly of vibration. The vibration signal is measured to detect the fault. Furthermore, it can analyze the magnitude of the vibration according to the rotation frequency to detect and diagnosis the cause of the fault [27]. In this chapter, the database was analyzed using the data-driven approach. By using the vibration data of the developed database, the characteristic change analysis was performed according to the cause of the fault.

Figure 4 shows fast Fourier transform (FFT) and short time Fourier transform (STFT) graphs of five pump fault modes. A raw signal is difficult to analyze as normal or fault. Therefore, by using FFT and STFT, the characteristic change according to the time domain and frequency domain was closely analyzed [28].

Figure 4a shows the normal signal as a graph. In the normal signal, amplitude changes of 1 mm/s$^2$ in the range of 2 k to 2.7 kHz were observed. Figure 4b shows the pump bearing a fault signal. The amplitude of the pump bearing fault signal was observed to increase to 2.8 mm/s$^2$ in the range of 2k~2.5 kHz, and it was also observed that the amplitude increased in the range of 3 k~4 kHz. Figure 4c shows a misalignment signal. It was observed that the amplitude of the fault signal increased to 4 mm/s$^2$ in the range of 0.9 k~1 kHz, and it was also observed that the amplitude increased in the range of 2.5k and 4.2 kHz. Figure 4d shows the jaw coupling fault signal. It was observed that the amplitude changes of the fault signal decreased compared to the normal signal in the range of 2 k to 2.7 kHz. This was because it was analyzed that the power transmission was decreased due to the occurrence of an elastomer degradation fault. Figure 4e shows a mechanical seal fault signal. It was observed that the amplitude of the fault signal increased to a maximum of 2 mm/s$^2$ in the range of 1.8 k to 2.5 kHz. Figure 4f shows the cavitation fault signal. The fault signal observed a continuous amplitude increase from the low frequency region to the range of 1.8 kHz.

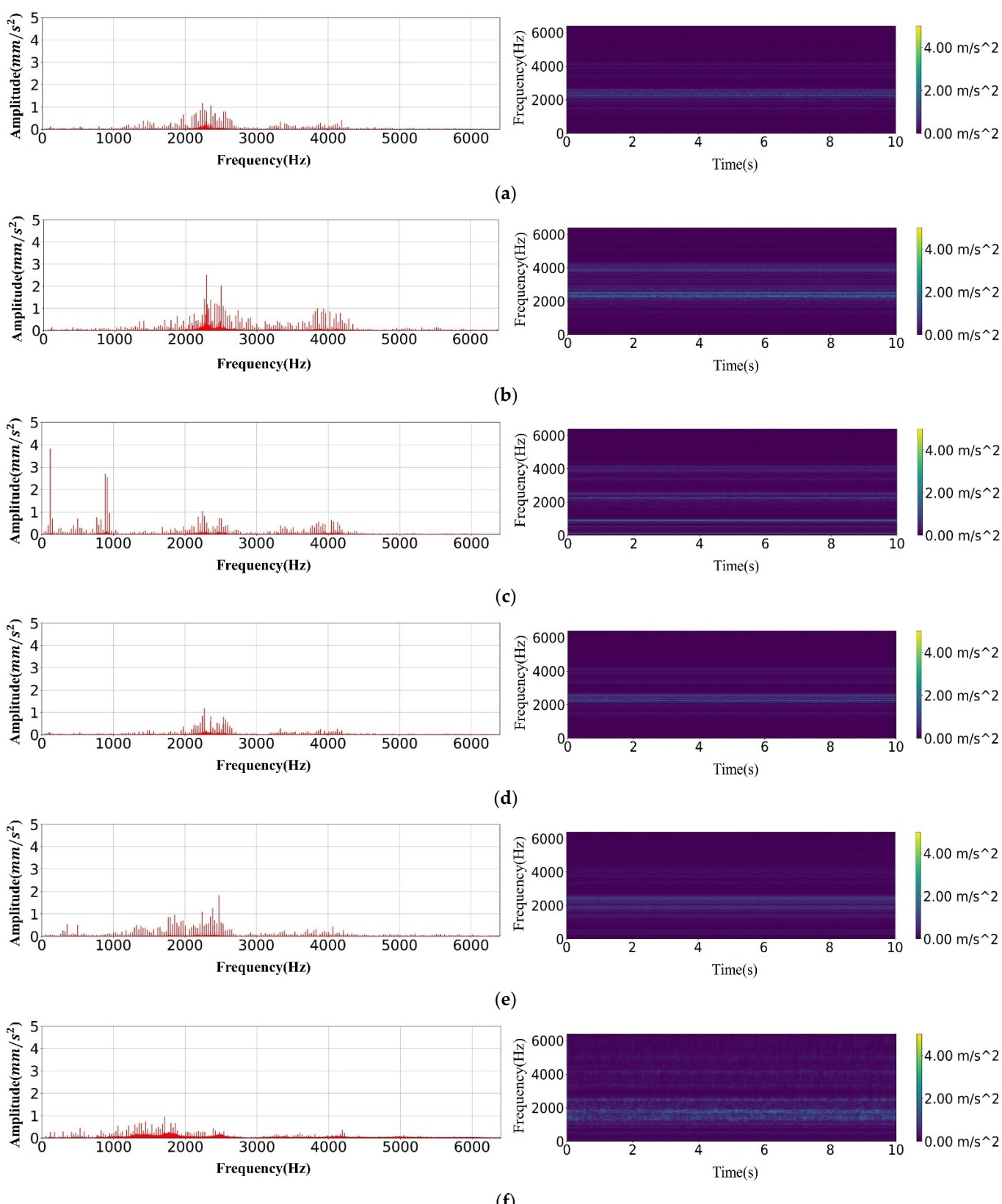

**Figure 4.** Graph of FFT and STFT of the vibration data: (**a**) Normal data; (**b**) Bearing particle injection test data; (**c**) Misalignment test data; (**d**) Jaw coupling elastomer test data; (**e**) Mechanical seal degradation test data; and (**f**) Cavitation test data.

## 4. Fault Diagnosis Based Deep Learning

In Section 3, the frequency analysis observed that there was a difference of the normal and fault. In this chapter, the data pre-processing proposed a method using time-series data without 2D image transformation. After that, the algorithm model was applied to the 1D CNN based on deep learning and evaluated the performance using a confusion matrix.

### 4.1. Data Preprocessing

The time-series data was composed of the sampling data points by extracting continuous values over time. The time-series data have various features. Furthermore, the time-series data have noise and a high dimensionality. For this, recently, there are various signal processing methods such as wavelet analysis and filtering. However, the signal processing method may require expert knowledge and may result in data information loss. Therefore, the time-series data can reduce the data pre-processing process such as wavelets and filtering [29].

The vibration data of the fault simulation database consist of the time-series data acquired for 10 min at a sampling rate of 25,600 Hz. The data used the vibration data x, y, z axes, and it was used without data feature extraction. In order to minimize the loss of the time series data having a one-dimensional array, the pre-processing process used the method of resampling to reduce the data size. Table 4 shows the data pre-processing process. In order to evaluate the fault classification performance according to the sampling rate of the data, the sampling rate of 25,600 Hz data was transformed into data having sampling rates of 12,800 Hz, 2560 Hz, 1280 Hz, 256 Hz, and 128 Hz. In addition, a hamming window was used to prevent data leakage errors. The data have different ranges of the feature according to the normal and fault modes, so we performed the normalization to convert them to a common scale. This pre-processing process was applied with the vibration data of the normal and five types of the fault modes, and labeling with values of 0 to 5.

**Table 4.** Resampling of pump vibration data.

| Sampling Rate (Hz) | Input Data | Columns | Window | Fault Modes | Labels |
| --- | --- | --- | --- | --- | --- |
| 25,600 | 84,480,000 | 3 (x, y, z) | Hamming | Normal | 0 |
| 12,800 | 42,240,000 | 3 (x, y, z) | Hamming | Bearing Fault | 1 |
| 2560 | 8,448,000 | 3 (x, y, z) | Hamming | Misalignment | 2 |
| 1280 | 4,224,000 | 3 (x, y, z) | Hamming | Jaw coupling | 3 |
| 256 | 844,800 | 3 (x, y, z) | Hamming | Mechanical seal | 4 |
| 128 | 422,400 | 3 (x, y, z) | Hamming | Cavitation | 5 |

### 4.2. CNN Model Construction and Training

CNN is one of the deep neural networks that is evaluated for high performance in image and video classification. It consists of several convolution and pulley layers, and outputs the data by weighted summing of the input data with a filter [30]. CNN models share the same feature regardless of the dimensions, but there is a difference in the dimension of the input data and the filtering approach. In general, 2D CNN models that learn the photos and videos learn features of the input data by filtering along the x and y axes. On the other hand, the 1D CNN model has a lower complicated calculation method than the 2D CNN by filtering only on the *x*-axis [31]. Due to the low computational processing requirements, it can be used in real-time and low-cost, even on low-performance computers [32].

Table 5 shows the structure of the 1D CNN model. Figure 5 shows an internal structure for the fault classification algorithm. It consists of four 1D convolution layers and one dense layer, and the activation functions were set as ReLU and soft max. It has a six-output node to classify input data into normal and five-types of failure modes. Max pooling was configured to form features between two 1D convolution layers and global average pooling

was configured to transform features into one-dimensional vectors. The drop out was set at 50% to minimize overfitting. The Adam optimizer was used to optimize the model, the learning rate was set to 0.001, the batch size was set to 100, and the epochs were set to 50. For data learning, train data was used for 80% of the input data and test data was used for 20%, and validation data was used for 20% of the train data.

**Table 5.** D CNN structure.

| Layer | Type | Filters Size | Filters Number | Activation |
|---|---|---|---|---|
| 1 | Convolution 1D | 100 | 15 | Relu |
| 2 | Convolution 1D | 100 | 15 | Relu |
| 3 | Max-pooling 1D | 100 | 1 | - |
| 4 | Convolution 1D | 100 | 10 | Relu |
| 5 | Convolution 1D | 100 | 10 | Relu |
| 6 | Global-average-pooling 1D | 100 | - | - |
| 7 | Dropout | 0.5 | - | - |
| 8 | Dense layer | 6 | - | Softmax |

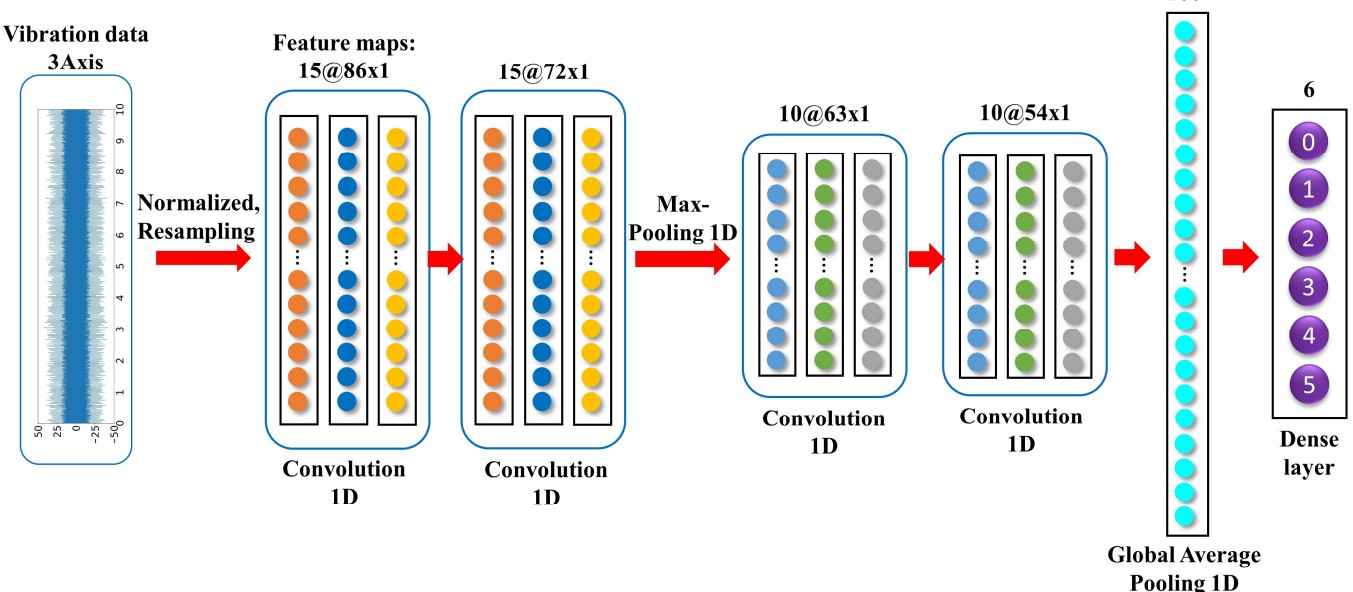

**Figure 5.** Internal structure of 1D CNN.

### 4.3. Evaluation Performance

The confusion matrix is one of the methods for representing the measured and predicted values according to the results of classification [33]. In addition, the confusion matrix visualizes and analyzes the distribution of predicted labels for measured labels in one table. This is the most useful method for the evaluation of the precision, recall, and F1-score. The evaluation for measured values and predicted values can be expressed as indices of the true positive (TP), true negative (TN), false positive (FP), and false negative (FN) [34]. Table 6 shows the confusion matrix, and Equation (1) calculates the precision through TP and FP. Equation (2) calculates the recall through TP and FN, and Equation (3) calculates the F1-score through TP, FP, and FN [35].

**Table 6.** Confusion matrix.

| | Predicted Positive | Predicted Negative |
|---|---|---|
| **Actual Positive** | TP | FN |
| **Actual Negative** | FP | TN |

The fault classification results were evaluated through the multi-class confusion matrix. The multi-class confusion matrix was composed of n × n dimensions with various class labels such as $A_0$, $A_1$, $\cdots$, $A_n$. Therefore, instances such as TP, TN, FP, and FN of a binary classification confusion matrix cannot be applied. However, based on this, it is possible to analyze with a focus on a specific class. It can be consisted as the multi-class confusion matrix by combining it as a whole [36,37]. The construction of the multi-class confusion matrix is shown in Table 7.

$$\text{Precision} \ = \ \frac{\text{TP}}{\text{TP} \ + \ \text{FP}} \tag{1}$$

$$\text{Recall} \ = \ \frac{\text{TP}}{\text{TP} \ + \ \text{FN}} \tag{2}$$

$$\text{F1 score} \ = \ \frac{2\text{TP}}{2\text{TP} \ + \ \text{FP} \ + \ \text{FN}} \tag{3}$$

**Table 7.** Multi-label confusion matrix.

| | | Predicted | | |
|---|---|---|---|---|
| | | **A₁** | **Aⱼ** | **Aₙ** |
| | **A₁** | $N_{11}$ | $N_{1j}$ | $N_{1n}$ |
| **Actual** | **Aᵢ** | $N_{i1}$ | $N_{ij}$ | $N_{in}$ |
| | **Aₙ** | $N_{n1}$ | $N_{nj}$ | $N_{nn}$ |

Table 8 shows the results of classification of the normal and fault data according to the sampling rate. Most of the input data were evaluated with the high performance of 0.99 or more in the accuracy, precision, recall, and f1-score. Figure 6 shows the accuracy and loss of train and validation data. Most of the input data were trained with a high accuracy of 0.9 or more, and the validation was also evaluated with the high accuracy. However, in the loss data, it was observed that the over-fitting occurred the lower the sampling rate of the input data, and this was analyzed as a lack of a data.

Figure 6a shows the accuracy and loss by a sampling rate of 25,600 Hz. Training and validation data were evaluated with a high accuracy of 1.00 and it was observed to have the low loss rate of less than 0.1. Figure 6b shows the accuracy and loss by a sampling rate of 12,800 Hz. This sampling rate was observed with the highest accuracy and it was observed to have a low loss rate of less than 0.01. Figure 6c shows the accuracy and loss by a sampling rate of 2560 Hz. This sampling rate was observed to have a higher accuracy than the sampling rate of 25,600 Hz, and the loss rate was also a low of less than 0.05. Figure 6d shows the accuracy and loss by the sampling rate of 1280 Hz. This sampling rate was observed as a high accuracy of 1.00 and it was observed to have a lower loss rate than the sampling rate of 12,800 Hz. Figure 6e shows the accuracy and loss by the sampling rate of 256 Hz. This sampling rate was observed to have a high accuracy, but the loss rate was observed as higher than another sampling rate. It was analyzed that the overfitting was caused by a lack of data. Figure 6f shows the accuracy and loss by a sampling rate of 128 Hz. This sampling rate was also observed with a high performance. In addition, the loss rate was observed to be as high as 0.2 or more. It was also analyzed that overfitting was caused by lack of the data. Figure 7 shows the fault classification results according to the sampling rate as a multi-class confusion matrix. This observed that the fault classification was classified with a high performance according to the measured and predicted values.

**Table 8.** Evaluation performance by sampling rate.

| Sampling Rate (Hz) | 25,600 | 12,800 | 2560 | 1280 | 256 | 128 |
|---|---|---|---|---|---|---|
| **Loss** | 0.00 | 0.00 | 0.00 | 0.00 | 0.00 | 0.0038 |
| **Precision** | 1.00 | 1.00 | 1.00 | 1.00 | 1.00 | 1.00 |
| **Recall** | 1.00 | 1.00 | 1.00 | 1.00 | 1.00 | 1.00 |
| **F1-score** | 1.00 | 1.00 | 1.00 | 1.00 | 1.00 | 1.00 |
| **Accuracy** | 1.00 | 1.00 | 1.00 | 1.00 | 1.00 | 0.9988 |

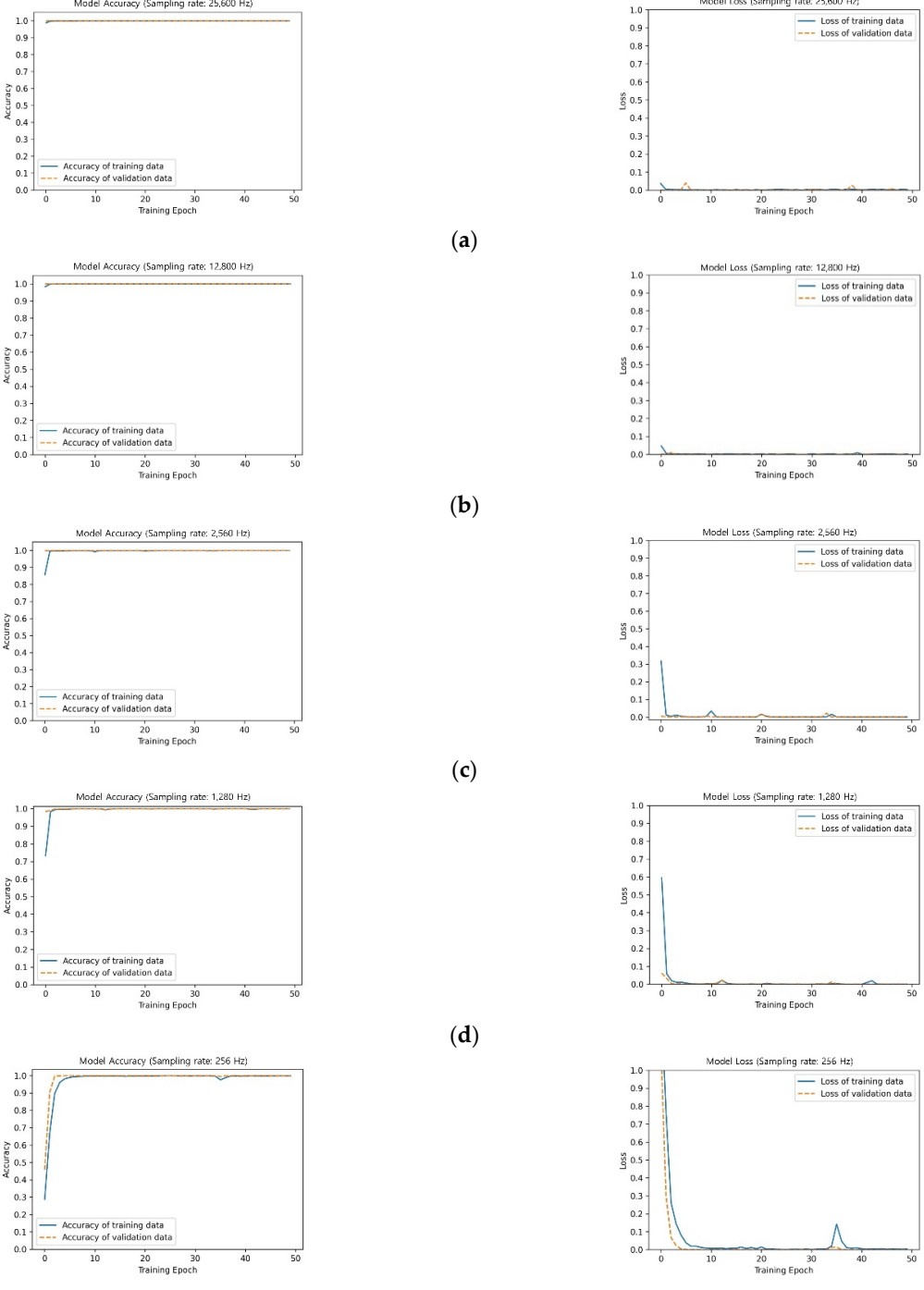

**Figure 6.** *Cont.*

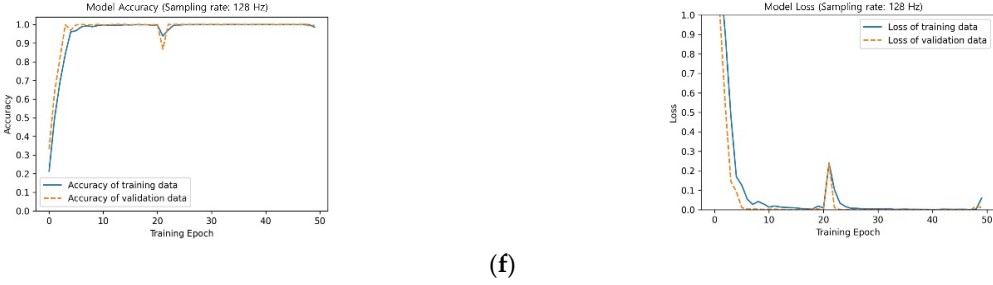

(**f**)

**Figure 6.** Accuracy and loss of the fault classification by sampling rate: (**a**) 25,600 Hz; (**b**) 12,800 Hz; (**c**) 2560 Hz; (**d**) 1280 Hz; (**e**) 256 Hz; and (**f**) 128 Hz.

(**a**)

(**b**)

(**c**)

(**d**)

(**e**)

(**f**)

**Figure 7.** Multi-class confusion matrix as the results of the fault classification by sampling rate: (**a**) 25,600 Hz; (**b**) 12,800 Hz; (**c**) 2560 Hz; (**d**) 1280 Hz; (**e**) 256 Hz; and (**f**) 128 Hz.

## 5. Conclusions

In this paper, a study on the development of the fault simulation database was proposed with various experimental methodologies. First, the reliability analysis has been performed to select five types of the core fault modes of the target device. A fault scenario that defines the fault simulation test methodology has been drawn based on several prior studies for the fault simulation tests of five types of fault modes. A land-based testbed has been built to perform the fault simulation test. The fault simulation database has been developed using a land-based testbed. Additionally, the faults were classified using a fault diagnosis algorithm based on deep learning. The performance of the fault classification was evaluated with a multi-class confusion matrix.

1 The reliability analysis was performed to select five types of the core fault modes such as bearing lubrication and wear, jaw coupling elastomer wear, mechanical seal degradation, misalignment, and cavitation. The fault simulation scenario was defined as the fault simulation test methodology of five types of the fault modes.

2 A land-based testbed was built to perform the fault simulation tests. The fault simulation database was constructed through the fault simulation test using a land-based testbed. The fuel supply system of the ships is integrated into several auxiliary pieces of equipment. However, it was built as an individual system to analyze the fault characteristics and performance of the target device.

3 The fault simulation test was performed on the bearing lubrication and particle injection, jaw coupling elastomer wear, mechanical seal degradation, misalignment, and cavitation test. The fault simulation database was developed with a total of 109 sets, which consisted of normal data of 5 sets, bearing fault of 20 sets, jaw coupling wear of 18 sets, mechanical seal degradation of 5 sets, misalignment of 32 sets, cavitation of 4 sets, and a bearing life test and complex fault test of 25 sets.

4 The frequency analysis of the constructed fault database was confirmed by validation to classify the normal and fault with a data driven approach. The frequency analysis was observed up to the maximum frequency range of data using FFT.

5 The fault classification was performed by applying a 1D CNN algorithm based on deep learning using the fault simulation database. The data pre-processing was performed using the resampling method to select the optimal sampling rate. The sampling rate was reduced with 25,600, 12,800, 2560, 1280, 256, and 128 Hz.

6 The performance of the fault classification according to the sampling rate was evaluated using a multi-class confusion matrix, and it was confirmed to have a high performance of the fault classification with 0.99 or higher on the accuracy and an F1-score of the low sampling rate.

This study will be used as basic data for the fault classification of pumps and can lead to a study to evaluate the residual life based on the mechanism of core parts. The constructed database will be expected to be the basis for the development of the fault diagnosis and prediction algorithms for MASS in the future. As a follow-up study, the complex test on the multi-fault modes will be performed to acquire the additional fault database. The constructed database will be verified reliably and validated based on the actual data of ships. This study will be the basis to establish an efficient maintenance plan for the various auxiliary equipment of ships, and it will be expected to have positive results in terms of the cost and health management technology.

**Author Contributions:** Conceptualization, T.-h.L. and Y.-j.K.; methodology, J.-y.K. and T.-h.L.; software, J.-y.K.; validation, J.-j.L. and W.-k.L.; experiments, J.-y.K., S.-h.L. and J.-j.L.; investigation, S.-h.L. and J.-j.L.; resources, T.-h.L. and Y.-j.K.; writing—original draft preparation, J.-y.K.; writing—review and editing, T.-h.L.; supervision, J.-w.P.; project administration, Y.-j.K.; funding acquisition, T.-h.L. All authors have read and agreed to the published version of the manuscript.

**Funding:** This work was supported by Autonomous Surface Ship project grant funded by Ministry of Trade, Industry and Energy (MOTIE) (No.20011164).

**Institutional Review Board Statement:** Not applicable.

**Informed Consent Statement:** Not applicable.

**Data Availability Statement:** Not applicable.

**Conflicts of Interest:** The authors declare no conflict of interest.

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
