# Peer review of "A Study on Deep Learning-Based Fault Diagnosis and Classification for Marine Engine System Auxiliary Equipment"

_processes, doi:10.3390/pr10071345_

Round 1

Reviewer 1 Report

In this work, experimental method studies were proposed to develop various data-bases based on reliability. First, reliability analysis has been performed to select 5 types of core fault modes of target device. Fault scenario that define the fault simulation test meth-odology has been drawn based on several prior studies for failure simulation tests of 5 types fault modes. A land based test-bed has been built to perform the fault simulation test. A database including various fault modes has been developed using a land based test bed. Additionally, Fault diagnosis algorithm based on deep learning was proposed. Per-formance of fault classification was evaluated as a confusion matrix.

My opinion is that the paper is well written and the results are described with details. Although these are not of high importance or originality I consider that the manuscript may be accepted for publication as a attempt to test-bed has been built to perform the fault simulation test. However, there are some remarks:

·      The authors have to shed light on the similarities and differences among their work and the literatures of the problem.  A clear explanation, what is the new result in their work, and how it is build up upon previous work in the field.

·      The introduction can be improved

·      The resolutions of figures are very bad with unclear legends and labels

·      The translation of figures should be included properly  

·      A conclusion section should be extended to include more details

In general, the authors have to describe in more detail the purpose of their study and its original contents.

If the authors submit a modified version according to my suggestions where they also give more details/explanations about the abovementioned criticisms, I could recommend the paper for publication.

Author Response

The authors would thank the detailed reviewer’s comments. Please see the attachment.

Reviewer 2 Report

1. Condition based maintenance should be discussed in a separate Section "methodology". 

2. In Keywords Section, ........ "base."  NOT "base;".

3. In Introduction Section, first line, put space between "Maritime autonomous surface ships" and its abbreviation (MASS).

4. Along the paper, please do comment 3, especially in introduction Section.

5. The motivations of this paper should be discussed in points in introduction Section.

6. What about the availability measure?. Discuss it.

7. What about the mean active lifetime and inactive lifetime?

8.  Why the authors did not use any statistical tool along the paper to prove the results?. 

9. Real data should be discussed if possible. 

10. What about the entropy measure?. Discuss. 

5. 

Author Response

(The authors gave the same response as above.)
